# Spatially-Aware U-Net to Defend against Adversarial Attacks on YOLO Object Detectors

## Abstract

Object detection models play a key role in self-driving systems where real-time high accuracy performance is crucial. While deep neural networks are performant, they are highly vulnerable to adversarial attacks which are subtle perturbations in the input image that induce detection errors. However, there are limited works on improving robustness of object detectors. While several diffusion-based methods have achieved strong defense against adversarial attacks, they often require a lot of computing resources and do not offer real-time inference speeds. This paper explores a defense mechanism to enhance the robustness of YOLO-based object detection systems against adversarial attacks in autonomous driving scenarios without modifying the underlying detector. The proposed defense is an attention U-Net augmented with spatial attention layers to enable the network to capture the salient information despite adversarial perturbations from the input. The training of our defender is guided by the defended detector to not only improve robustness against adversarial examples but also maintain performance when evaluating benign images. Benchmarking against previous defense approaches, our method achieves thorough protection with less damage to the original detection ability. Moreover, our defense model exhibits a generalized defensive ability against various unseen adversarial attacks.

## 1 Introduction

Object detection is a crucial task in computer vision, essential for various applications such as autonomous driving Feng et al. (2021), robotics Xu et al. (2022), and surveillance Mishra & Saroha (2016). Various object detectors have been proposed in recent years. Specifically, You Only Look Once (YOLO) models Redmon et al. (2016) have gained popularity due to their effectiveness in accurate real-time visual recognition. Notwithstanding their prowesses, these models are vulnerable to adversarial examples Szegedy et al. (2013) designed to fool them Im Choi & Tian (2022), raising significant concerns. Depending on the available information about the victim model, adversarial attacks can be categorized into white-box and black-box attacks. In a white-box attacking scenario, the attacker has full knowledge of the victim model's architecture, weights and gradients. Whereas in a black-box setting, the attacker does not have access to that kind of information. Adversarial attacks can be further classified into targeted and untargeted attacks. Targeted attacks aim at making the model not detect an object or misclassify it as a specific incorrect category. Untargeted attacks simply try to cause a failure in detection without a target.

In order to increase the robustness of image classification models against adversarial attacks, adversarial training Kurakin et al. (2016); Tramèr et al. (2017) was naturally first introduced. Adversarial training (AT) refers to using adversarial examples while training a deep neural network. Another strategy is to remove the adversarial perturbations from the adversarial input. Leveraging this idea, diffusion-based models Nie et al. (2022) have achieved state of the art (SOTA) defending performance. Conversely, very few methods for defending object detection models have emerged. In order to solve this issue, a naive idea is to use existing defense methods for image classifiers for defending object detectors. While diffusion-based models offer better defense than AT, they are not suitable for real-time applications.

To address these issues, we introduce a U-Net based network augmented with spatial attention Woo et al. (2018) layers to remove adversarial noise. During training, the defense network is guided with

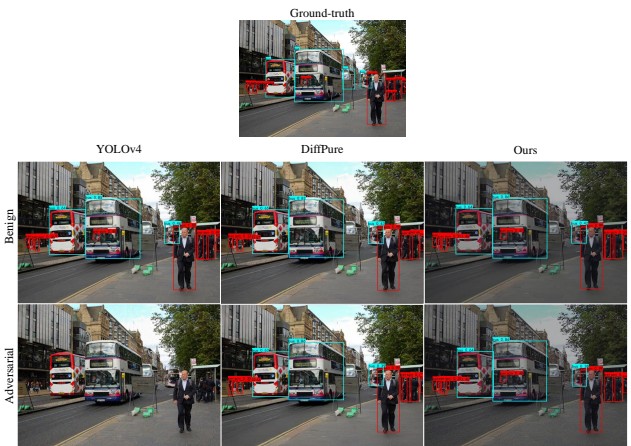

Figure 1: Detection results comparison of a YOLOv4 model without defense, with diffusion (Diff-Pure) defense and our defense. The detections are shown for a benign and an adversarial example.

the YOLO detector's loss function in order to maximize the defended YOLO's detection performance. Furthermore, the set of training images is augmented with adversarial examples to train the network to deal with adversarial images. The results of our defense against DiffPure (a diffusion-based model) is shown in Figure 1. As can be seen, using our defense allows the detector to still detect the bus driver whereas the diffusion-based one cannot. Overall, our defense method achieves more than 20% mAP@50% on some attacks compared to existing defense methods while having only around a 3.5% mAP@50% performance drop on benign (clean) images.

To summarize, the main contributions of this work are as follows:

- We propose a spatially-aware U-Net (SAU-Net) for removing adversarial noise in victim images. It is trained using a task-driven training strategy towards object detection.
- We compare our method with existing defense methods under SOTA adversarial attacks and show our model's increased robustness.

## 2    RELATED WORK

### 2.1    OBJECT DETECTION

The objective of object detection is to identify and categorize objects that are present in an image and to locate them with rectangular bounding boxes to indicate the degree of certainty that they exist. Deep learning-based object detection methods can be divided into two categories: two-stage and one-stage methods. R-CNN Girshick et al. (2014), Faster R-CNN Girshick (2015) and Mask R-CNN He et al. (2017) are two-stage methods that generate region proposals then classify and regress bouding boxes. One-stage detectors on the other hand, predict bounding boxes and class probabilities in one step. YOLO models Redmon et al. (2016) are one-stage detectors known for their real-time accurate processing capabilities. In this paper, we focus on defending adversarial attacks on the YOLO-based model.

### 2.2    ADVERSARIAL ATTACK METHODS

**Adversarial attacks on image classifiers.**  For classification, the Fast Gradient Sign Method (FGSM) attack Goodfellow et al. (2015) is well-known for its simplicity. It is a single-step attack that updates the input image in the direction to maximize the classification loss. The Projected Gradient Descent (PGD) attack Madry et al. (2019) improves upon the FGSM attack by taking multiple gradient update steps. By adding momentum, the Momentum-Iterative FGSM attack Dong et al. (2018) further enhance the PGD attack. Other attacks such as the Jacobian-based Saliency Map Attack Papernot et al. (2016), Carlini-Wagner (CW) attack Carlini & Wagner (2017) or DeepFool attack Moosavi-Dezfooli et al. (2016) have been thoroughly researched.

**Adversarial attacks on object detectors.** One of the first attacks crafted against object detectors is the Dense Adversary Generation (DAG) attack Xie et al. (2017). However, the DAG attack can only fool proposal based detectors and it requires high computation cost. Thus, Wei et al. propose the United and Efficient Adversary (UEA) attack which uses a Generative Adversarial Network (GAN) to generate adversarial examples. UEA can successfully fool both regression and proposal based detectors and can be used in video object detection. PGD attacks can be extended to object detection Zhang & Wang (2019) by leveraging the object detector's loss function. Chow et al. have developed the Targeted Objectness Gradient (TOG) attack which can be seen as a targeted PGD attack. For time-efficient attacks, Li et al. have generated universal adversarial perturbations that have been trained to attack any image. By exploiting Non-Maximum Suppression (NMS), the Daedalus attack Wang et al. (2021) increases the false positive rate in detection results while the UAP attack Shapira et al. (2023) increases the inference time of the detector. More recently, strong transferable black-box attacks Cai et al. (2022; 2023); Ding et al. (2024) have been developed.

### 2.3 Defense Against Aversarial Attacks

**Defense for image classifiers.** In order to defend against adversarial attacks targeted at classifiers, one of the ways is to use adversarial training Wong et al. (2020); Bai et al. (2021). By noticing that adversarial perturbations get larger in the feature space, Xie et al. have added feature denoising blocks using non-local means operation in the classifiers which they train end-to-end with adversarial training. However, adversarial training leads to a trade-off between standard and adversarial accuracy Madry et al. (2019). Thus the GAN-based method defense-GAN Samangouei et al. (2018) and the autoencoder-based method MagNet Meng & Chen (2017) aim at modeling the latent distribution of the standard images in order to move the adversarial images closer to it. However, apart from adversarial training, the aforementioned methods only use MNIST and/or CIFAR Krizhevsky et al. (2009) datasets which have low resolution images. Liao et al. use a U-Net for defense which is guided during training by high-level information from the classifier they want to protect. As random noise can attenuate adversarial perturbations Li et al. (2019), Wu et al. use an adaptive Wiener filter to filter out both the added noise and the adversarial perturbations. Currently, diffusion based models Nie et al. (2022); Wang et al. (2022); Zhang et al. (2024) have become the state of the art when it comes to defending against adversarial attacks. However, the major drawback of diffusion models is their long inference time.

**Defense for object detectors.** Defense against object detection has mostly been studied through adversarial training Zhang & Wang (2019); Chen et al. (2021); Im Choi & Tian (2022). While gabor convolutional layers Amirkhani & Karimi (2022) have been explored for defending against adversarial attacks, adversarial training remains the current state of the art.

## 3 Proposed Method

In this section, we introduce our defense strategy for defending against adversarial attacks for YOLO object detectors. The goal is to effectively "denoise" adversarial samples while minimizing the damage on clean samples using a UNet and attention. Therefore, we utilize both clean and adversarial examples to train our defense model.

### 3.1 Problem Formulation

For a clean image $x$, we denote $x_{adv}$ an adversarial example generated by a function $f_{adv}$. We assume that the attacker can have complete access to the detector $\mathcal{D}$'s model structure, weights, gradients, loss $\mathcal{L}_{\mathcal{D}}$ and outputs. Thus, an adversarial example is computed as following:

$$x_{adv} = f_{adv}(x, y, \mathcal{D}, \mathcal{L}_{\mathcal{D}}) \qquad (1)$$

In our setup, the attacker has black-box knowledge about the defender. Therefore, when computing a gradient-based attack (FGSM for example) it can only leverage the gradients of the detector $\mathcal{D}$. This gray-box scenario is illustrated in Figure 2. Our objective is to design and train a defender $f_{def}$ that can mitigate the impact of adversarial images in this gray-box scenario. Moreover, our defender should not degrade too much the detector's performance on clean images.

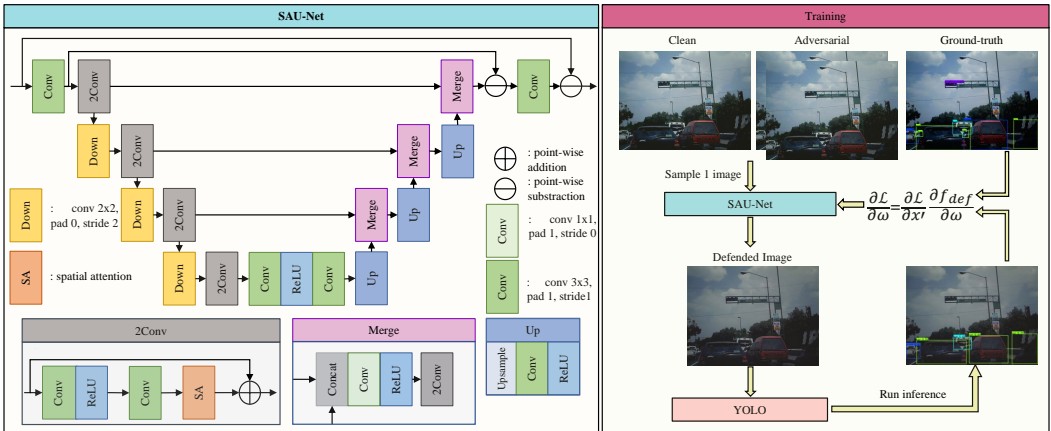

Figure 2: Defense strategy in a gray-box attacking scenario. The input to the defender can be either an adversarial or a benign image.

Figure 3: Defender network architecture (left) and training strategy (right). The defender is a 3-level deep spatially-aware U-Net. The training of the defender is geared towards the detection ability of the underlying YOLO detector.

## 3.2 DEFENDER ARCHITECTURE

Our defense strategy is illustrated in Figure 3. It is based on two main components:

- A spatially-aware U-Net to better process the adversarial noise present in the input image.

- A training strategy utilizing the detector's loss function to "purify" potential adversarial noise.

**Spatially-aware U-Net.** We employ a modified 3-level U-Net in our defense architecture. Instead of using fully-convolutional up and down blocks, we add spatial attention modules Woo et al. (2018) to help the network capture salient information despite the adversarial perturbations. As Xie et al. have discovered, the adversarial noise present in the input image is transferred into the feature maps. The architecture of our spatial attention modules is the same as Woo et al.'s. Spatial attention is computed and applied as such where $\oplus$ denotes the concatenation operation.

$$att_x = \sigma(conv_{ReLU}(maxpool(x) \oplus avgpool(x))) \tag{2}$$

$$y = x \times att_x \tag{3}$$

Thus, our down blocks are composed of two convolutional layers with ReLU activation in between and the spatial attention module as shown in the *2Conv* block in Figure 3. A skip connection is also added to increase the learning ability of the block He et al. (2016). For downsampling, we employ a 2×2 convolution layer with a stride of 2. In the up blocks, for upsampling, we use nearest-neighbor interpolation and a convolutional layer with ReLU activation. For combining the feature maps from the skip connection and the upsampled ones, concatenation is used followed. 1×1 convolution and ReLU activation function are used to reduce the number channels. Afterwards, a convolutional block augmented with spatial attention (like in the down blocks) follows. We employ residual learning in order to learn the adversarial noise.

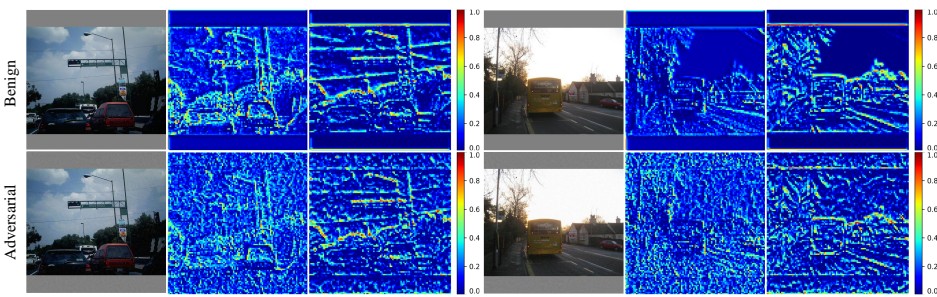

Figure 4: Feature map visualization after the first two stages of the YOLOv4 backbone for benign (top) and adversarial (bottom) images.

### 3.3 TRAINING STRATEGY

When defending against adversarial attacks, one idea is to move the adversarial image closer to its clean counterpart. For that, the most intuitive idea is to train a defender $f_{def}$ using a distance loss between the defended input and the clean input.

However, this method is not directly geared at improving the detection performance under adversarial attacks. Therefore, the approach of guiding our defense network in a pixel-wise manner may not be the most effective approach as it would require the defender to almost completely remove the adversarial noise. As one can notice in Figure 4, the feature maps can get really noisy for adversarial examples. Liao et al. have thus proposed using a distance loss between the high-level feature maps obtained from the clean and adversarial inputs. Alternatively, they also use the classification loss for training their defense. Inspired by this, we train our defender using the detector's loss function $\mathcal{L}_{\mathcal{D}}$. By doing so, the defender is trained in a task-driven manner for object detection. YOLOv4's Bochkovskiy et al. (2020) loss function is composed of 3 parts:

$$\mathcal{L}_{\mathcal{D}} = \lambda_{obj}\mathcal{L}_{obj} + \lambda_{loc}\mathcal{L}_{loc} + \lambda_{cls}\mathcal{L}_{cls} \tag{4}$$

where $\lambda_{obj}$, $\lambda_{loc}$ and $\lambda_{cls}$ represent the weight assigned their respective loss components.

In order to preserve performance on clean images while defending against adversarial ones, the defender is trained on both clean and adversarial examples. For a clean input:

$$\mathcal{L}_{clean}(x,y) = \mathcal{L}_{\mathcal{D}}(f_{def}(x), y) \tag{5}$$

For an adversarial input:

$$\mathcal{L}_{adv}(x,y) = \mathcal{L}_{\mathcal{D}}(f_{def}(f_{adv}(x,y)), y) \tag{6}$$

Then, for defending against $n_{adv}$ attacks, with $\mathcal{L}_{adv}^i$ denoting the loss against attack number $i$:

$$\mathcal{L}_{adv} = \sum_{i=1}^{n_{adv}} \lambda_i \cdot \mathcal{L}_{adv}^i \tag{7}$$

$$\text{with } (\lambda_i)_{i \in \{1,...,n_{adv}\}} \in [0,1]^{n_{adv}}$$

Finally, if we denote $\lambda_{clean} \in [0,1]$ the weight assigned to the loss stemming from clean inputs:

$$\mathcal{L} = \lambda_{clean} \cdot \mathcal{L}_{clean} + \mathcal{L}_{adv}$$

$$\text{with } \lambda_{clean} + \sum_{i=1}^{n_{adv}} \lambda_i = 1 \tag{8}$$

Therefore, if we denote $\omega$ as the defending network's weights, the optimization objective becomes:

$$\min_{\omega} \mathcal{L}(x,y;\omega) \tag{9}$$

Let $f_{def}(x;\omega) = x'$, the loss gradients with respect to the defender's weights $\omega$ are:

$$\frac{\partial \mathcal{L}}{\partial \omega} = \frac{\partial \mathcal{L}}{\partial x'} \frac{\partial f_{def}}{\partial \omega} \tag{10}$$

In practice, during training, on each iteration, a clean or adversarial example is randomly sampled from the $\lambda$ distribution.

## 3.4 DATASET

The experiments are carried out on the KITTI Geiger et al. (2012) and COCO traffic datasets. The categories of the KITTI dataset are combined like in Im Choi & Tian (2022). It is composed of three categories: car, cyclist and pedestrian. We use the original training images (7,481 images) and split them half into a training set and the other half into a testing set. The COCO traffic dataset is a subset of the MS-COCO Lin et al. (2014) dataset. It contains 8 categories: person, bicycle, car, motorcycle, bus, truck, traffic light and stop sign. The training set of the COCO traffic dataset is the traffic subset (71,536 images) of the 2017 MS-COCO training set. In a similar manner, we collect our testing set by utilizing the traffic subset (3,028 images) of the 2017 MS-COCO validation set.

## 3.5 EVALUATION METRICS

In order to compare various defense methods, we use two metrics: mAP@0.5 for pure performance and FPS for inference speed.

**mAP@0.5.** The defense models are evaluated using the Pascal VOC mean average precision metric. We use an IOU threshold of 0.5 (mAP@0.5). The clean mAP refers to the mAP@0.5 evaluated on clean inputs. Whereas, the robust mAP will refer to the mAP@0.5 evaluated on adversarial examples.

**FPS.** For evaluating the inference speed of the defense models, we use the frames per second (FPS) metric. Since the inference time baseline is the YOLO detector's, FPS will be measured as the inference speed of both the defense and YOLO networks combined. All the inference speed tests are conducted on a single Tesla P100 GPU and on a fixed subset of 500 randomly selected images for each dataset.

## 3.6 YOLO DETECTOR

The detector used in our experiments to train the defender is YOLOv4 Bochkovskiy et al. (2020). During the training of the proposed defender, the detector's weights are kept frozen. We also use YOLOv8/11 for testing in order to evalutate our defender's generalization capabilities.

## 3.7 ATTACK BASELINE

We evaluate our proposed defense method against a variety of attacking methods including **FGSM** Goodfellow et al. (2015), **PGD** Madry et al. (2019), **MI-FGSM** Dong et al. (2018), TOG-vanishing (**TOG-van**) Chow et al. (2020), TOG-fabrication (**TOG-fab**) Chow et al. (2020), Carlini-Wagner (**CW**) Carlini & Wagner (2017) and Universal Dense Object Suppression (**U-DOS**) Li et al. (2021). The parameters of the attacks are described in Appendix B.1.

## 3.8 DEFENSE BASELINE

Our proposed defense method is evaluated against 4 different defense methods. Adversarial training (**AT**) is evaluated using the results presented in Im Choi & Tian (2022). A simple image denoising method is evaluated with the Adaptive Wiener Filter (**AWF**) Wu et al. (2020). By default, we add gaussian noise with $\mu = 0$ and $\sigma = 0.2$ to the input image. In order to get a thorough comparison against denoising methods, **SUNet** Fan et al. (2022) is also tested. We use the official pretrained weights provided by the authors. Gaussian noise is again added with $\mu = 0$ and $\sigma = 0.2$. For the diffusion model, we select **DiffPure** Nie et al. (2022).

# 4 EXPERIMENTS

## 4.1 RESULTS

**Robustness analysis.**

**YOLOv4.** (Table 1) On the KITTI dataset, the defense method has better robust mAP than the other defenders on all attacks but the CW attack. Moreover, SAU-Net can maintain accuracy drop

Table 1: Comparison with various defense methods against different attacks on YOLOv4. The best defense result for each attack is highlighted in **bold**. A result in parentheses means that the AT model was used for evalutation.

| | mAP@0.5 ↑ (%) | | | | | | | | | |
| | KITTI | | | | | COCO traffic | | | | |
| | None | AWF | SUnet | DiffPure | Ours | None | AWF | SUnet | DiffPure | Ours |
|---|---|---|---|---|---|---|---|---|---|---|
| Benign | **73.09** | 57.57 | 62.8 | 66.38 | 69.25 | **62.18** | 55.24 | 59.19 | 57.87 | 58.99 |
| FGSM | 19.59 | 41.36 | 43.77 | 46.86 | **66.59** | 23.53 | 41.25 | 40.17 | 44.27 | **58.76** |
| FGSM8 | 18.69 | 30.23 | 29.32 | 33.68 | **58.89** | 24.42 | 35.85 | 36.71 | 30.88 | **53.21** |
| PGD | 0.72 (49.43) | 45.31 | 51.1 | 55.66 | **67.40** | 1.55 (34.77) | 44.97 | 49.87 | 53.04 | **57.64** |
| PGD8 | 0.22 | 32.74 | 41.92 | 45.93 | **61.41** | 1.05 | 41.94 | 44.84 | 47.32 | **54.44** |
| TOG-van | 0.81 | 46.96 | 56.39 | 62.39 | **63.81** | 2.54 | 46.5 | 54.61 | **56.17** | 55.38 |
| TOG-fab | 14.14 | 55.90 | 61.06 | 63.63 | **64.43** | 19.25 | 53.85 | **57.87** | 56.65 | 55.74 |
| MI-FGSM | 1.84 | 38.37 | 41.06 | 44.25 | **66.45** | 1.42 | 37.43 | 38.75 | 45.8 | **58.24** |
| MI-FGSM8 | 0.56 | 21.86 | 20.29 | 24.22 | **58.84** | 0.73 | 31.02 | 22.16 | 33.64 | **53.03** |
| U-DOS | 16.93 | 56.52 | 63.54 | 64.68 | **68.14** | 12.55 | 54.74 | **58.56** | 58.42 | 57.88 |
| CW | 58.86 | 56.22 | 62.06 | **65.23** | 63.87 | 44.49 | 53.53 | 56.78 | 57.42 | **59.25** |

Table 2: Comparison with various defense methods against different attacks against YOLOv8 on the KITTI dataset. The best defense result for each attack is highlighted in **bold**.

| | mAP@0.5 ↑ (%) | | | | | | | | | |
| | KITTI | | | | | COCO traffic | | | | |
| | None | AWF | SUnet | DiffPure | Ours | None | AWF | SUnet | DiffPure | Ours |
|---|---|---|---|---|---|---|---|---|---|---|
| Benign | **83.5** | 65.7 | 72.5 | 80.0 | 77.4 | **66.5** | 61 | 63.5 | 63.6 | 62.4 |
| FGSM | 46.6 | 50.3 | 60 | 65.8 | **68.8** | 38.2 | 51.5 | 53.5 | 55.7 | **57.4** |
| FGSM8 | 35.6 | 38.3 | 44.8 | 52.0 | **53.7** | 34.5 | 45.1 | 44.9 | 45.1 | **50.1** |
| PGD | 17.6 | 47.9 | 60.5 | 71.2 | **71.6** | 20.8 | 53.9 | 57.5 | **58.9** | 58.7 |
| PGD8 | 4.7 | 32.2 | 49.2 | 58.5 | **60.5** | 6.44 | 47.2 | 51.2 | **54.4** | 52.6 |
| TOG-van | 28.0 | 52.7 | 65.5 | **73.3** | 68.0 | 32.2 | 57.8 | 61.2 | **61.6** | 57.7 |
| TOG-fab | 49.3 | 63.9 | 72.6 | **74.9** | 71.9 | 30.3 | 58.1 | 60.5 | **61.6** | 58.4 |
| MI-FGSM | 19.1 | 41.2 | 54.6 | 61.3 | **66.6** | 14.0 | 49.4 | 51.8 | 55.1 | **57.1** |
| MI-FGSM8 | 8.94 | 22.4 | 30.3 | 33.0 | **46.1** | 5.58 | 35.4 | 35.5 | 38.2 | **47.8** |

within 10% (compared with clean mAP) under the most of the attacks. That is a considerable improvement compared to previous defense methods which can have robust mAP drops of more than 20%. The defense method demonstrates a high level of robustness against strong attacks such as MI-FGSM and MI-FGSM8 and outperforms DiffPure by approximately **22%** and **34.6%**, respectively. Furthermore, on benign images, the mAP drop is only of **3.84%** with SAU-Net compared to using a standard YOLO model. This is a slight upgrade relative to the already decent performance of DiffPure in that area.

On the COCO traffic dataset, the overall performance of the proposed defense relative to the clean YOLO is pretty similar to KITTI. However, there are some noticeable differences. The robust mAP decline is barely discernible in comparison to the clean mAP for untargeted attacks. Also, the defense model is more robust to untargeted attacks using a higher attack budget ($\epsilon = 8/255$) compared to the KITTI dataset. However, for targeted attacks (TOG-van, TOG-fab, U-DOS), our defense model is a bit less robust than previous defenses. Overall, our defense model can generalize to defense against various unseen targeted and untargeted attacks. This is further corroborated by the fact that our defender was trained solely on untargeted attacks. This fact might also explain some of the small performance drop against targeted attacks when comparing to other defense methods. A visual comparison displaying the detections of the various defense methods under adversarial attacks can be found in Figure 5.

**YOLOv8.** The first observation (Table 2) that can be made is that the defender trained on YOLOv4 can be used to defend YOLOv8. Compared to the defenseless YOLOv8 model, the defended YOLOv8 model is drastically more adversarially robust. The improved robustness can be of up to 55.8% on the PGD8 attack for example. Nonetheless, this comes at the cost of a performance drop when evluating against clean images of 6.1% for the KITTI dataset and 4.1% for the COCO traffic dataset. Another trend that was also observed with YOLOv4 is that the proposed defender

Table 3: Comparison with various defense methods against different attacks against YOLOv11 on the KITTI dataset. The best defense result for each attack is highlighted in **bold**.

| | mAP@0.5 ↑ (%) | | | | | | | | | |
| | KITTI | | | | | COCO traffic | | | | |
| | None | AWF | SUnet | DiffPure | Ours | None | AWF | SUnet | DiffPure | Ours |
|---|---|---|---|---|---|---|---|---|---|---|
| Benign | **82.3** | 57.1 | 68.0 | 78.4 | 67.7 | **67.3** | 61.0 | 64.1 | 64.3 | 62.6 |
| FGSM | 44.9 | 45.6 | 57.1 | **65.2** | 58.0 | 41.1 | 53.3 | 56.4 | 58.0 | **58.3** |
| FGSM8 | 34.3 | 36.0 | 45.0 | **51.8** | 41.4 | 37.2 | 46.6 | 48.6 | 48.7 | **52.0** |
| PGD | 15.5 | 40.9 | 57.2 | **69.9** | 59.2 | 22.0 | 55.1 | 59.7 | **60.1** | 59.3 |
| PGD8 | 4.30 | 29.0 | 46.9 | **60.4** | 45.7 | 7.54 | 49.2 | 54.2 | **56.6** | 54.6 |
| TOG-van | 29.1 | 47.0 | 60.8 | **73.0** | 53.8 | 36.5 | 58.5 | 62.7 | **62.8** | 57.4 |
| TOG-fab | 44.0 | 56.0 | 69.3 | **74.5** | 57.1 | 31.3 | 59.0 | 62.4 | **62.9** | 58.9 |
| MI-FGSM | 16.8 | 36.3 | 50.3 | **61.3** | 55.3 | 15.7 | 51.1 | 54.9 | 57.3 | **57.9** |
| MI-FGSM8 | 6.63 | 19.3 | 30.6 | 35.7 | **36.2** | 6.33 | 38.8 | 41.4 | 43.2 | **51.1** |

| | KITTI | | COCO traffic | |
| | $t \downarrow$ (ms) | FPS ↑ | $t \downarrow$ (ms) | FPS ↑ |
|---|---|---|---|---|
| YOLOv4 | 21.0 | 47.62 | 20.8 | 48.01 |
| AWF | 22.02 | 45.41 | 21.87 | 45.72 |
| DiffPure | 2950 | 0.34 | 3862.8 | 0.26 |
| Ours | 26.09 | 38.33 | 25.92 | 38.58 |

Table 4: Average inference time (ms) for a single image and FPS (defense + YOLOv4).

performs almost always better on untargeted attacks than the other defenders on both the KITTI and COCO traffic datasets. However, for targeted attacks, DiffPure offers better robustness. Another interesting fact is that the larger the perturbation budget gets, the better SAU-Net performs when compared to other defense methods. This was also present for YOLOv4. Therefore, it seems that YOLOv8 and YOLOv4 share core features that the defender exploits to improve their adversarial robustness

**YOLOv11.** For YOLOv11 (Table 3), the robustness transfer is done to a lesser extent. That is to be expected as it is vastly different than YOLOv4 in terms of architecture. Notably, YOLOv11 has attention layers. Therefore, the attacks computed on YOLOv11 differ enough that the trained defender on YOLOv4 has less adversarial robustness. That fact is especially apparent in the KITTI dataset where the SAU-Net cannot keep up with the diffusion model in terms of performance. On the COCO traffic dataset, the SAU-Net provides satisfactory results that are on par with DiffPure on untargeted attacks. The most plausible explanation of the performance drop on the KITTI dataset may be that the defender overfit the YOLOv4 model. As a matter of fact, the KITTI dataset is quite small and all of its images are of the same size. Therefore, the defender might have overfit due to this.

**Inference speed.** In autonomous driving systems, real-time inference of images is crucial. Therefore, in order to better evaluate the defense models from different aspects, we also compare their inference speeds in Table 4. Our defense model (including YOLO inference) can be processed at around 38.46 FPS. This is a decrease compared to the 47.62 FPS achieved by simply only using a YOLO detector. However, real-time inference is still very feasible. Filter-based method like AWF introduces minimal impact to inference speed. However, it has the worst defense performance compared with others. Contrasting with our model, using a diffusion-based defense degrades the processing speed to around 0.3 FPS. Similarly, SUnet also necessitates a greater inference duration than our model. Consequently, it is impractical to employ these two approaches for autonomous driving applications that demand real-time computing. Our solution provides a good balance between defense and speed.

## 4.2 ABLATION STUDY

We investigate the impact of different elements of our model architecture in Table 5. In order to evaluate the impact of the spatial attention layers, we remove them in the U-Net and denote this

Table 5: Comparison of various model architectures against the FGSM, PGD and MI-FGSM attacks.

| | mAP@0.5 ↑ (%) | | | |
| | | KITTI | | |
| | benign | FGSM | PGD | MI-FGSM |
|---|---|---|---|---|
| U/0 | 68.98 | 66.24 | 67.11 | 66.28 |
| SU/0 (ours) | **69.25** | 66.59 | **67.40** | **66.45** |
| CU/0 | 68.35 | 66.30 | 66.35 | 65.37 |
| SCU/0 | 68.49 | 65.82 | 65.56 | 65.17 |
| U/1 | 68.03 | 65.42 | 65.78 | 65.13 |
| SU/1 | 67.99 | 65.84 | 66.42 | 65.57 |
| CU/1 | 68.27 | **66.77** | 66.18 | 65.9 |
| SCU/1 | 67.99 | 65.20 | 65.89 | 64.86 |

model as U/0. This model is essentially analogous to Liao et al.'s defensive approach in image classification. As our observation, adding the spatial attention layers (SU/0) seems to improve both the accuracy and robustness of the underlying YOLO model. Consequently, we may consider incorporating channel attention blocks after the SAU-Net to enhance the network's performance. This may assist the network in selecting the most relevant channel-specific information to provide to the detector.

A channel attention block is composed of a channel attention module Hu et al. (2018), a convolution layer for reducing the number of channels and a LeakyReLU activation. For an input $x$ with multiple channels, the channel attention module will compute channel attention weights $att_x$ for an input $x$. Then, for each channel, the corresponding attention weight is applied. For $\sigma$ as the sigmoid function:

$$att_x = conv_\sigma(conv_{ReLU}(avgpool(x))) \qquad (11)$$

$$y = x \times att_x \qquad (12)$$

This block should allow the defender to only choose the relevant information for the object detector. We denote the number of channel attention (CA) layers as such $/n_{ca}$ in Table 5. However, adding channel attention layers seems to hamper the network's ability to deal with both clean and adversarial examples. This is even more noticeable with the MI-FGSM attack which the defender has not seen during training.

Another idea is to add channel attention modules as in the original attention U-Net Oktay et al. (2018). This is denoted as CU. However, it shows no noticeable performance gain when increasing the number of CA layers. Overall, when comparing U/, SU/ and CU/, the U-Net with only spatial attention (SU/) performs better than the others. This suggests that spatial attention layers are key to the network's performance. However, combining both spatial and channel attention in the U-Net seems to bottleneck the network's performance. For structure simplicity and effectiveness to attacks, we only integrates spatial attention to the final model structure.

## 5 CONCLUSION

In this work, a spatially-aware U-Net is designed to defend against adversarial attacks against YOLO object detection models in a gray-box attacking scenario. To capture both salient information and potential adversarial noise, we employ spatial attention to extract weighed spatial information. The proposed model is optimized to maximize the performance of the underlying object detector by learning to "purify" the input image without requirement of modifying or fine-tuning the detector itself. Extensive experiments show the proposed method can enhance the robustness of YOLO detector under various attacks. Besides, the proposed method has lightweight model structure and fast processing speed compared with other SOTAs. Therefore, our method can be deployed for the application that requires real-time processing such as self-driving system. For future work, it can be interesting to investigate the scenario where the attacker has white-box knowledge of the defense method through developing a self-supervised learning framework to learn adversarially robust representations.

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

# A APPENDIX

# B EXPERIMENTAL DETAILS

## B.1 ADVERSARIAL ATTACKS PARAMETERS

For the FGSM attack, we use an attack budget of $\epsilon = 4/255$ and $\epsilon = 8/255$ (FGSM8). The PGD attack is evaluated with an attack budget of $\epsilon = 4/255$, a step size of $\alpha = 1/255$ and 10 iterations. A more powerful variant of PGD (PGD8) is used with $\epsilon = 8/255$ and $\alpha = 2/255$. We also use the MI-FGSM attack with an attack budget of $\epsilon = 4/255$, a step size of $\alpha = 1/255$, a decay factor of $\mu = 1$ and 10 iterations. Similar to PGD8, MI-FGSM8 is also tested. The TOG-van and TOG-fab attacks are evaluated with an attack budget of $\epsilon = 8/255$, a step size of $\alpha = 2/255$ and 10 iterations. The CW attack is adapted to object detection using the detector's loss function in an untargeted setting. The number of iterations is set to 50 and the learning is set to 0.01. Unless specified otherwise, all the untargeted attacks (FGSM, PGD, MI-FGSM, CW) mentioned above generate their attacks using

| $\lambda_{clean}$ | KITTI mAP@0.5 ↑ (%) | | | | | | |
|---|---|---|---|---|---|---|---|
| | 0 | 0.1 | 0.2 | 0.3 | 0.4 | 0.5 | 0.6 |
| Benign | 68.80 | 69.25 | 68.63 | 68.79 | 68.79 | **69.66** | 69.61 |
| FGSM | **66.95** | 66.59 | 66.18 | 65.70 | 64.63 | 63.58 | 62.70 |
| PGD | 66.42 | **67.40** | 66.12 | 65.93 | 65.63 | 65.28 | 64.78 |
| MI-FGSM | 65.96 | **66.45** | 65.10 | 64.84 | 64.02 | 62.87 | 61.76 |

Table 6: Comparison between various $\lambda_{clean}$ used to train the SAU-Net. For each attack, the best result is highlighted in **bold**. $\lambda_{FGSM}$ and $\lambda_{PGD}$ are always equal to $\frac{1-\lambda_{clean}}{2}$. Overall, $\lambda_{clean}$ provides the best results when considering both clean and adversarial robustness.

the objectness loss of the detector. This is done because Im Choi & Tian have shown that these types of attacks are the most effective against YOLO detectors. The U-DOS attack is generated using the same training procedure as specified in the original paper.

### B.2 IMPLEMENTATION DETAILS

We use Python 3.9 and Pytorch 2.0 with CUDA 11.7 for the implementation of our defense method. The models are trained using a single Tesla P100 or V100 GPU and the Adam optimizer.

For training, we use a combination of benign, FGSM and PGD images. The adversarial images are generated based on the training dataset. The FGSM and PGD attacks used here have the same parameters as specified in the Attack Baseline section. We set $\lambda_{clean} = 0.1$ and $\lambda_{FGSM} = \lambda_{PGD} = 0.45$ as they provide the best results as shown in Table 6. For the KITTI dataset, the initial learning rate is set to 0.001 and decays by 0.9 every epoch. The batch size is 4 and the model is trained for 40 epochs. For the COCO traffic dataset, the initial learning rate is set to 0.0004 and decays by a factor of 0.9 every epoch. The batch size is 12 and the model is trained for 50 epochs. For both datasets, the Adam optimizer is used.

## C VISUAL COMPARISON

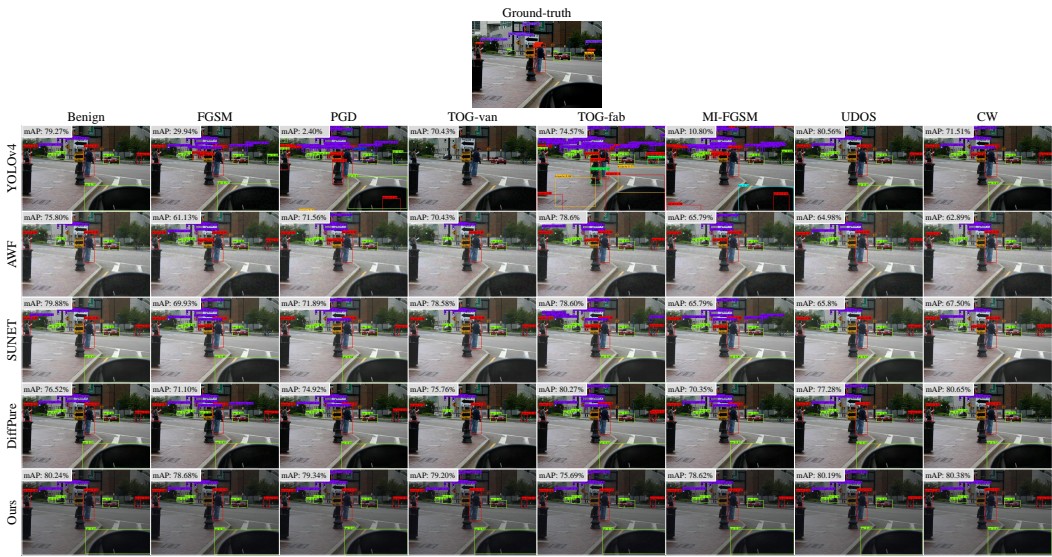

Figure 5: Detection result comparison between various defense methods (left) under an array of attacks (up) on the COCO traffic dataset.

