# OpenReview forum: "Spatially-Aware U-Net to Defend against Adversarial Attacks on YOLO Object Detectors"
_ICLR.cc/2026/Conference — Submitted to ICLR 2026_

### Official Review · Reviewer_BpBV · 2025-10-25

**Soundness:** 2
**Presentation:** 2
**Contribution:** 1
**Rating:** 2
**Confidence:** 4

**Summary:**

This paper proposes a defense to adversarial attacks computed against YOLO systems, in feature space (i.e. by formulating attacks that works only in the pixel space). The defense works by training a model to apply de-noising on samples, achieved at training time by learning the removal of specific noise patterns. According to the experiments, the methodology seems better than regular de-noise methods and adversarial training.

**Strengths:**

+ I agree, there is less that is achieved for defending systems like YOLO (which are widely more deployed than other models for the image domain)
+ Interesting de-noising technique which leverage previously-computed adversarial example
+ Many attacks are tested against models and defenses trained on different datasets

**Weaknesses:**

This paper is interesting, but its contributions are too narrow for the scope of this conference.
Also, there are technical flaws that erodes the contribution proposed by this paper.

**1) Attacks are badly setup.** When computing the robustness of a system, an attacker tries their best to evade detection. Here the attacks are setup with very few iterations and a super small stepsize: I wonder whether the failure of attacks is due to this bad setup. Also all iterative attacks could be wrapped together (PGD, MI-FGSM) since they perform the same actions but with few difefrences (normalization of gradients, norm, etc). Hence, results should be similar, which is not the case in the presented numbers. The authors should better and carefully check these, understand and better discuss what is going on.

**2) Adversarial training?** How this method has been setup? The paper is obscure to this respect and it is unclear how models were hardened with this simple technique. Also, in many settings, DiffPure is a real competitor, thus showing that there is no clear winner since attack strategies are all mixed up in terms of iterations, stepsize, norm.

**3) No comparison in terms of computations.** This method also uses training samples to compute adversarial examples, fed to the de-noiser. However, no complexity is being reported when compared with adversarial training. Also, AT models are for sure faster, since they are the same YOLO model but hardened.

**4) No adaptive evaluation.** This de-noising strategy is end-to-end differentiable, and worst-case evaluation could show that a slighlty tuned PGD can break it by just differentiating w.r.t. the perturbation. This is not discussed in the paper, but it should.

**5) Bad presentation.** The figure at the beginning of the paper is impossible to read, and the formatting is odd (subsections with just few lines of text occupying a lot of space with no reason).

**References**

Tramer, F., Carlini, N., Brendel, W., & Madry, A. (2020). On adaptive attacks to adversarial example defenses. Advances in neural information processing systems, 33, 1633-1645.

Pintor, M., Demetrio, L., Sotgiu, A., Demontis, A., Carlini, N., Biggio, B., & Roli, F. (2022). Indicators of attack failure: Debugging and improving optimization of adversarial examples. Advances in Neural Information Processing Systems, 35, 23063-23076.

**Questions:**

1) Can you report the results of experiments showing that attacks are correctly setup?
2) Is this methodology really better than simple adversarial training?

---

### Official Review · Reviewer_KEmn · 2025-10-29

**Soundness:** 2
**Presentation:** 2
**Contribution:** 2
**Rating:** 2
**Confidence:** 4

**Summary:**

This paper proposes a spatially-aware U-Net (SAU-Net) as a purifier to defend YOLO object detectors against adversarial attacks. The method integrates spatial attention and trains the purifier using the detector’s loss. Experiments are conducted on KITTI and a COCO “traffic” subset, showing improved robustness under several attacks while maintaining near real-time speed.

**Strengths:**

Robustness of object detectors under adversarial attacks is an important and less‐studied dimension compared to classification models.

**Weaknesses:**

1The paper assumes a weak/grey-box attacker: the adversary does not adapt to the purifier. Modern adversarial defense literature expects adaptive attacks that consider the full pipeline (purifier + detector). Without this evaluation, the claimed robustness is likely overstated.

2 The method shows degraded robustness on YOLOv11, indicating poor generalization across detector architectures.

3 No evaluation on larger or more diverse datasets, or against patch-based attacks, which limits practical applicability.

4 Evaluated solely on YOLO-based detectors with no evidence of effectiveness on transformer-based or other detection architectures, limiting generalizability.

**Questions:**

1 Have you evaluated the SAU-Net purifier under adaptive attacks where the adversary has knowledge of both the detector and the purifier (end-to-end attack)?

2 Your method is evaluated only on YOLO-based detectors. How would it perform on transformer-based detectors (e.g., DETR, Swin-Transformer) or other architectures such as Faster R-CNN or RetinaNet?

3 Experiments are limited to KITTI and a COCO “traffic” subset. How would the method perform on larger-scale or more diverse datasets (full COCO, Open Images Dataset)?

4 Why does the paper not consider patch-based attacks, which are more realistic and relevant for real-world applications?

---

### Official Review · Reviewer_CKuk · 2025-10-31

**Soundness:** 2
**Presentation:** 3
**Contribution:** 2
**Rating:** 4
**Confidence:** 4

**Summary:**

This paper proposes a Spatially-Aware U-Net (SAU-Net) defense mechanism to protect YOLO object detectors against adversarial attacks in autonomous driving scenarios.

**Strengths:**

Clear Practical Motivation

- Addresses an important problem: adversarial robustness of object detection in autonomous driving
- Clearly identifies the inference speed bottleneck of existing diffusion-based methods (~0.3 FPS cannot meet real-time requirements)

Reasonable Method Design

- Spatial attention mechanism is theoretically motivated for capturing salient information and filtering adversarial perturbations
- Using the detector's loss function for task-driven training is a sensible design choice
- Residual learning strategy is appropriate for noise learning tasks

**Weaknesses:**

1. Missing Adaptive Attacks

The most serious weakness is the absence of adaptive attack evaluation, which is now considered essential in adversarial defense research [Tramer et al. 2020, Croce & Hein 2020]

Evidence of the problem:

Table 1 shows without defense: PGD achieves 0.72% mAP, but with SAU-Net defense: 67.40% mAP
This dramatic "improvement" suggests the gray-box assumption severely weakens the attack
The attacker optimizes perturbations against the detector only, unaware of the defender, so the defender can "fix" the adversarial image

Why this matters:

In real deployment, attackers can reverse-engineer or gain knowledge of defense mechanisms
Gray-box evaluation provides an overly optimistic assessment of robustness
The defense might be easily bypassed with adaptive attacks

What's needed:

White-box evaluation where PGD/C&W attack through the full pipeline (defender + detector)
Evaluation against standard adaptive attack suites (e.g., AutoAttack)
At minimum, PGD with end-to-end backpropagation through both defender and detector
Discussion of computational overhead for such attacks

Mitigation: This could be partially addressed by showing that even when attacking the defender+detector jointly, the high computational cost makes it impractical for real-time scenarios.

2. Limited Technical Novelty

U-Net for adversarial defense: Not novel (Liao et al. 2018 already proposed similar approach for classification). The paper acknowledges this but positions SAU-Net as essentially the same architecture (U/0 in Table 5).
Spatial attention: Direct application of existing CBAM work (Woo et al. 2018)
Task-driven training: Conceptually similar to Liao et al.'s high-level representation guided denoiser
Main contribution: Combining these existing techniques for YOLO defense with empirical validation

The novelty is primarily in the application domain and empirical findings rather than methodological innovation.
Quantitative evidence: Table 5 shows SAU-Net (SU/0) improves over vanilla U-Net (U/0) by only ~0.3-0.5% mAP on most attacks, suggesting limited benefit from the spatial attention component.

3. Generalization Concerns

YOLOv11 Performance Drop (Table 3):
On KITTI: SAU-Net underperforms DiffPure on 6 out of 9 attacks (notably PGD8: 45.7 vs 60.4, TOG attacks: 53.8 vs 73.0)
Authors attribute this to "overfitting" due to small KITTI dataset, but this explanation is concerning:

If the method overfits to YOLOv4's specific architecture, it limits practical applicability
The small dataset excuse doesn't explain why performance on COCO traffic (larger dataset) also shows similar patterns



Deeper concern: The defender may be learning YOLOv4-specific features rather than general adversarial perturbation patterns. This is evidenced by better generalization from YOLOv4→YOLOv8 (more similar architectures) than YOLOv4→YOLOv11 (more different).
Suggested experiments:

Training on multiple detector types
Analysis of what features SAU-Net learns
Explicit architectural compatibility study

4. Training-Testing Data Leakage

The defender is trained on FGSM and PGD, then evaluated on these same attacks. While the paper also tests on unseen attacks (TOG, MI-FGSM, CW, U-DOS), the overlap makes it difficult to assess true generalization:

Seen attacks (FGSM, PGD): Results may be optimistic
Unseen attacks: Should be clearly separated in results tables
Better protocol: Train on subset of attacks, report results on completely held-out attack types

Table 1 interpretation: The strong performance on MI-FGSM (trained) vs. TOG-van (unseen) suggests some generalization, but this needs clearer analysis.

5. Insufficient Technical Details for Reproducibility

Training Details:

Why λ_clean = 0.1? Table 6 shows λ_clean = 0.5 achieves better robust mAP on some attacks
What is the training time and GPU memory requirement?
How are FGSM/PGD adversarial examples generated during training? (Same ε as testing?)
Data augmentation strategy?

Defense Deployment:

How much does SAU-Net increase end-to-end latency compared to bare detector?
What is the memory footprint?

6. Incomplete Analysis and Understanding

a) Failure Cases Not Analyzed:

Why does SAU-Net underperform on CW attack (Table 1: 63.87 vs 65.23 for DiffPure)?
Why does performance on targeted attacks (TOG) vary significantly between datasets?
Figure 1 may show a cherry-picked example - systematic failure analysis is needed

b) Defense Mechanism Not Well Understood:

What exactly does SAU-Net learn? Feature map visualization (Fig. 4) is insufficient
How do defended images differ from clean images? (PSNR/SSIM/LPIPS metrics needed)
Visualization of spatial attention maps would provide insight into what the network focuses on
Does the defender primarily remove high-frequency noise? Spectral analysis would help

c) Channel Attention Failure Unexplained:

Table 5 shows channel attention (CU/) and combined (SCU/) perform worse than spatial-only (SU/)
This contradicts findings from CBAM and other attention work
Possible explanations: implementation issue, incompatibility with residual learning, or task-specific finding?
Deeper investigation needed

7. Experimental Limitations

a) Loss Function Design:

Only uses detector loss L_D, no image quality constraints
This may lead to unnatural-looking defended images
Should include perceptual loss or adversarial training-style regularization
Need metrics: PSNR, SSIM, LPIPS to assess visual quality

b) Attack Strength:

PGD with only 10 iterations is relatively weak (standard: 20-100)
ε = 4/255 and 8/255 are moderate budgets; should test ε = 16/255
Missing stronger attacks: AutoAttack, FAB, Square Attack
CW attack parameters not fully specified (confidence parameter?)

c) Baseline Comparisons:

AT results from different paper (Im Choi & Tian 2022) - unclear if same experimental setup
DiffPure designed for classification - did authors optimize hyperparameters for detection?
SUNet is for image denoising, not adversarial defense - somewhat unfair comparison

8. Writing Issues

a) Notation Inconsistency:

Equation (1): f_adv takes L_D as input, but under gray-box assumption, attacker shouldn't access defender information. Is this L_D of detector or combined system? Clarify.
Section 3.1 description could be more precise

b) Overclaimed Statements:

Abstract: "thorough protection" is too strong given YOLOv11 results
"Generalized defensive ability" overstates findings
Should be more measured in claims

**Questions:**

1. Adaptive Attack Evaluation (Critical)

Can you provide evaluation results where the attacker has white-box access to both the defender and detector?
Specifically, please report the robust mAP when PGD attack is performed with end-to-end backpropagation through the full pipeline. Table 1 shows that PGD achieves only 0.72\% mAP without defense but 67.40\% with SAU-Net, suggesting your gray-box evaluation significantly weakens the attack. This white-box evaluation is essential to assess the true robustness of your defense, as attackers in deployment can reverse-engineer the defense mechanism.

2. Cross-Detector Generalization Analysis

Table 3 shows significant performance degradation on YOLOv11, particularly on KITTI dataset (e.g., PGD8: 45.7\% vs 60.4\% for DiffPure, TOG-van: 53.8\% vs 73.0\%). You attribute this to overfitting, but can you provide deeper analysis: (a) What specific architectural differences between YOLOv4 and YOLOv11 (e.g., YOLOv11's attention layers) might explain this gap? (b) Can you visualize what features or patterns SAU-Net learns that are YOLOv4-specific versus generalizable? (c) If you fine-tune SAU-Net on YOLOv11, how much does performance improve? This is crucial for understanding whether your defense can be practically applied to new detector architectures.

3. Training-Testing Overlap and True Generalization

Your defender is trained on FGSM and PGD attacks, which are also in your test set. Can you report results separately for: (a) Seen attacks (trained on): FGSM, PGD and their variants; (b) **Unseen attacks** (not trained on): TOG-van, TOG-fab, MI-FGSM, CW, U-DOS. Additionally, what is the performance if you train only on PGD and evaluate on all other attacks? This separation is essential to assess true generalization capability versus memorization of training attack patterns.

4. Technical Novelty and Marginal Gains

Table 5 shows that your SAU-Net (SU/0) improves over vanilla U-Net (U/0, analogous to Liao et al.'s method) by only 0.3-0.5\% mAP on most attacks. Given this modest improvement: (a) Can you provide the exact computational overhead (FLOPs, parameters, latency) introduced by spatial attention modules? (b) Can you justify why this marginal gain represents a significant contribution beyond Liao et al.'s approach? (c) Why does channel attention (CU/0, SCU/0) hurt performance, contradicting findings from CBAM and related work? Is this an implementation issue or a fundamental finding?

---

### Official Review · Reviewer_WuF8 · 2025-11-01

**Soundness:** 2
**Presentation:** 2
**Contribution:** 2
**Rating:** 2
**Confidence:** 3

**Summary:**

This paper introduces a spatially-aware U-Net (SAU-Net) as a defense mechanism to improve the robustness of YOLO-based object detectors against adversarial attacks. The proposed method augments a U-Net with spatial attention layers and is trained using both clean and adversarial examples, guided by the YOLO detector’s loss function. Extensive experiments on KITTI and COCO traffic datasets show that SAU-Net outperforms existing defense methods in terms of both adversarial robustness and inference speed, making it suitable for real-time applications.

**Strengths:**

1. This paper proposes a spatially aware U-Net variant with spatial attention layers that are designed to remove adversarial noise in images. This model is tailored for object detection pipelines like YOLO and does not require modifications on the YOLO model. The defender is trained using the loss function of the YOLO detector itself, thus ensuring the performance of the detector after purification. Thus, the method shows resilience against both targeted and untargeted attacks. The method also showcases better inference time compared to the diffusion-based methods, making it practical in real-life scenarios like automated driving.
2. The architecture and training methodology are clearly stated and explained in the paper. The paper shows better performance of the defense method under multiple attacks compared to the baseline.

**Weaknesses:**

1. While the paper demonstrates that the spatial attention layers help in denoising the input image, they do not provide theoretical proof as to why it happens. The whole architecture appears to be empirically motivated rather than theoretically derived. The core idea of the method is to use a spatially spatial attention module in U-Net, but the method just applies a combination of methods without any theoretical insights into the mechanism.
2. In my point of view, this paper is just an incremental engineering as opposed to a theoretical innovation. They just used already known components such as U-Net backbone and spatial attention modules, which are well established. The novelty of the method lies solely in the combination of these components. The ablation experiments shown in (Table 5) shows that a U-Net without a spatial attention layer works almost as good as this one, thus implying that the benefit from the spatial attention components are marginal.
3. The scope of the paper is narrowed down due its application only in a gray-box scenario. This study fails to evaluate against the adaptive attacks which are aware of the defender’s architecture. It makes this defense method susceptible to gradient based attacks where the attacker could optimize against the outputs of SAU-Net. The claim of robustness is not strong enough. This paper misses evaluation against adaptive attacks, thus failing to prove the robustness against stronger attack methods. The evaluation also focuses only on two datasets – KITTI and COCO – which are narrow in scope. Both datasets include traffic scenarios, limiting the diversity of the evaluation. The defense method is not tested against other domains, creating uncertainty about generalization to broad real-world settings.
4. The paper also lacks good presentation with a lot of scope for improvement in terms of writing and presentation. There are numerous errors in grammar and wording. In addition, there are technical inconsistencies in figures and mathematical notations. For example, in Figure 3, conv:1x1 is shown to use stride 0 which doesn’t make any sense. Another notable example is the inconsistent use of f_adv() function. This function is dependent on detector model according to equation (1), but loses that dependency when it comes to equation (6).

**Questions:**

1. The paper mentions that the attacker was trained on YOLOv4. The paper also shows results against YOLOv8 and YOLOv11. Was the attacker retrained on YOLOv8 and YOLOv11 on that scenario? If not, won’t that have any impact on the effectiveness of the attack on the latter experiments?
2. The method assumes the attacker has access to the architecture of the detector while being unaware of the architecture of the defender. How realistic is this setting in the real-world systems? Wouldn’t the attacker who can optimize to the detector also optimize to the defender?

---

### Meta-Review · Area_Chair_s4yK · 2026-01-07

**Summary:**

None of the reviewers suggested accepting the paper and there is no rebuttal. The main concerns are lack of sufficient evaluation (eg. limited to YOLO models only) and limited novelty.

**Reviewer Concerns:**

The main concerns reported by reviewers are: it evaluates robustness only under a weak gray-box threat model and omits adaptive (white-box) attacks, making the reported robustness gains likely overstated and potentially due to weak attack configurations. The technical contribution is largely incremental, relying on well-known components without providing theoretical justification, while ablation results show only marginal benefits from the proposed attention module. The evaluation scope is narrow, limited to traffic datasets and YOLO-based detectors, with degraded performance on newer architectures indicating poor generalization. Additional concerns include overlapping training and testing attacks, insufficient attack strength, lack of computational cost analysis, and poor presentation.

**Reviewer Scores:**

2,4,2,2

There was no rebuttal.

---

### Decision · Program_Chairs · 2026-01-26

Reject